# Contribution of Particles to Air Pollution in Green Parks

Jelena S. Kiurski [1], Vladimir M. Rajs [2,*] , Snežana M. Aksentijević [3], Aleksandra M. Čavić [1]
and Dragan D. Soleša [1]

1   Faculty of Economics and Engineering Management in Novi Sad, University Business Academy in Novi Sad,
    Cvećarska 2, 21000 Novi Sad, Serbia; jelena.kiurski7@gmail.com (J.S.K.);
    aleksandra.cavic22@gmail.com (A.M.Č.); dragan.solesa@fimek.edu.rs (D.D.S.)
2   Department of Power, Electronic and Telecommunication Engineering, Faculty of Technical Sciences,
    University of Novi Sad, Trg Dositeja Obradovića 6, 21000 Novi Sad, Serbia
3   Western Serbia Academy of Applied Studies, Užice Department—College of Applied Sciences,
    Trg Svetog Save 34, 31000 Užice, Serbia; sneza.aksentijevic@gmail.com
*   Correspondence: vladimir@uns.ac.rs

**Abstract:** Parks can aid in the regulation of microclimates and the improvement of air quality. They can be utilized in real-world systems to choose the best model for explaining the source of pollutant emissions, indicating the requirement for pollution concentration monitoring. Monitoring concentration trends is critical to formulating a strategy to reduce $CO_2$ emissions and the contribution of these gasses to the greenhouse effect, as well as to curbing the rising levels of PM in the air. The research background of this study was performed in the green parks of Novi Sad, Serbia. The results are represented in terms of the quantity of the pollutants, and the correlation of the examined phenomena through statistical analysis. Aeroqual monitors with laser sensors were used to take measurements of particle pollution ($PM_{2.5/10}$). The constant was confirmed by inter-comparison laboratory measurements of air-quality quantitative control. The measurement findings revealed a minor variance in concentration values for $PM_{2.5/10}$ from 26–30 μg/m$^3$, which were within the allowed limits, indicating that the air was moderately clean. The linear link between particle concentrations and nitrogen dioxide in the sample was also validated using simple linear regression, as was the high influence of humidity on particle concentrations.

**Keywords:** ambient air; particle pollution; green parks; city of Novi Sad; inter-comparison laboratory measurements; simple regression analysis

## 1. Introduction

Today, a large part of mankind lives in big cities under modified climate conditions. Many of the climate modifications that are caused by big cities have a negative impact on human health, such as, for example, reduced airflow, which also contributes to rising temperatures and air pollution. Characteristics of cities that lead to unfavorable local climate conditions can be fixed with appropriate planning measures in the up-building or the reconstruction of a city. At the same time, bearing in mind that even though different cities have many mutual characteristics, each individual city, and even parts of the same city, have their own climate details [1–3]. The important sources of air pollution are larger industrial facilities, i.e., industrial zones, and heating systems. The assessment of the urban environment is performed based on the quality of the environment, the spatial–functional structure of the city, the evaluation of space from an ecological aspect, the evaluation of space from the aspect of health, etc.

The goal of measuring air pollution is to take preventive measures in all segments, such as: examination of the influence of polluted air on people's health, nature, and material goods; monitoring concentration trends; reviewing the impact of measures taken on the level of air pollution; and informing the public. In the implementation of the environmental protection system, all individual and legal subjects are responsible for the

failure to implement environmental protection measures, in accordance with the law (The Law on Environmental Protection) [4]. Protecting clean air is achieved by taking measures such as systematic monitoring of air quality, reducing air pollution and keeping pollutants below the prescribed limit values, taking technical, technological, and other necessary measures to reduce emissions, and monitoring the impact of polluted air on human health and the environment [4]. Measurement methods, equipment, tested parameters, as well as interpretation of results should be followed under the Rule Book on limit values, emission (concentration level is a quantitative measure of emission) measurement methods, criteria for establishing, measuring points, and data records [1,5].

The city of Novi Sad is the administrative center of the South Bačka district and of the Autonomous Province of Vojvodina and is located on the riverside of the Danube [6]. Novi Sad has an exceptional natural-geographic location, which is endorsed by the fact that it is right on the crossroads of waterways and roadways. It is located on the eastern tourist route which connects Northern Europe, Central Europe, and West Europe with the Adriatic Sea, the Aegean Sea, and the Black Sea. The importance of this route, as well as Novi Sad, is emphasized even more thanks to the Pan-European Corridor VII (the river Danube) and the Pan-European Corridor X (Central Europe—Black Sea and Aegean Sea), which have a direct influence on this area. The city is positioned on the main European E-75 road, which represents the main artery for Southern, Central and Eastern Europe. Besides, this intersection is located about 50 km from the international road E-70, which connects Western and Eastern Europe.

Novi Sad can be considered an economically developed and dynamic city, which achieves significant economic prosperity, thanks to the fact that there are important energy, petrochemical and chemical plants located on its territory. Unfortunately, it just so happens that the existing economy, primarily industrial, drove Novi Sad to be one of the most endangered cities in Serbia from an environmental standpoint. There are several causes for this condition, and those causes can, according to the document environmental protection [1,7], be classified into four groups:

1. Causes originating from the development of the socialist economy, industrialization and up-building, from the fifties to the seventies in the 20th century.
2. Causes that are linked to the period of the economic downturn and the political chaos that occurred in the nineties in the 20th century.
3. Causes that were induced by the consequences of the NATO campaign in 1999.
4. Causes that arose after the year 2000, as a result of the institutional wandering and the discontinuity in the work of state bodies and institutions in the environmental areas.

From the air pollution aspect, traffic and the prominent morphological characteristics of the city are the main sources of air pollution. Besides traffic, high concentrations of pollutants in the air are caused by the street's width, height, and the layout of buildings, as well as the inclination of the streets. Air quality is indirectly influenced by the relief in certain parts of the city. Considering that the geomorphologic composition of the territory of Novi Sad is made out of silt-sized matter (loess and sand), their transmission in the air is possible [8]. Novi Sad used to be a greener city, but today, with the upbuilding of modern housing zones, it came down to having only 10 parks. In 2019, Novi Sad joined the EBRD (European Bank for Reconstruction and Development) Green Cities with a vision to improve green infrastructure and ensure a sustainable future for the city and its inhabitants [9].

The total protective role of greenery on a city's territory is dependent on the quantity, type, and quality of the greenery, but also on the layout and the quality of the established relationships with the greenery in its surroundings. The positions of existing organized and partially organized parks do not cover all of the zones in the city equally, such as the Danube Park in the center of the city which covers only a small territory. Residents of Liman have the most favorable area. Aside from having the walking path next to the Danube, they can also use the partially arranged unities. Futoški Park is a park with a

special purpose around the Iodine SPA, and it is protected as a natural monument, as a natural good of the II and III categories [10].

However, an increase in pollutant levels in the air has been observed not only in Serbia, but throughout Europe, in the form of fine suspended particles ($PM_{2.5}$; PM—particulate matter), which are emitted during the process of coal and biomass combustion (in households, commercial and industrial plants), as well as coarse suspended particles ($PM_{10}$), which come from industry and transportation and are emitted directly into the air [11,12]. Suspended particles represent the most dangerous form of air pollution, given that they have the ability to penetrate deep into the lungs and bloodstream, and in doing so they can cause cancer, mutations of DNA, heart attacks, and premature death [13,14].

The numerous consequences of air pollution, including the death of one million people per year on a global scale, are frequently the primary concern of the domestic public. Exposure to air pollution from outside air is linked to a large number of acute and chronic diseases, from irritation to respiratory and cardiovascular diseases [15]. The effects of air pollution on public health are well documented, although the mixture of various types of pollutants in the air can be complex. Air pollution is made up of a mix of liquid and solid phases; mixtures of gaseous, volatile, semi-volatile, and suspended particles with a very variable ratio [16]. The main pollutants studied for their effects on health are suspended particles, ozone, nitrogen dioxide, sulfur dioxide, methane, mercury, and soot from the combustion of hydrocarbons. The concentration of industrial plants, located in one place, out-of-date technologies, dense traffic, and non-compliance with the laws regarding environmental protection affect the quality of green parks as well as zones for leisure and recreation of children and adults. Recreation, in a general anthropological sense, represents a special activity for residents, which adds to the development and preservation of physical and mental health, vitality, quality of life, rest, and pastime, all the things that can relax, revitalize, and provide respite for people today. This points out the growth in popularity of places for so-called active recovery. In an urban environment, zones for resting and recreation are of exceptional importance. Given that the residents of urban zones often do not have too much free time to leave the urban zones, park zones are a daily haven for children as well as adults [17]. Because parks are often within housing projects, besides the roads or important intersections, ambient air has a big impact, which in Novi Sad is quite polluted and as such has an effect on the health of children and adults.

The agglomeration of "Novi Sad" covers the territory of the city of Novi Sad. With the regulation on monitoring conditions and demands for air quality [18], limit values are issued for sulfur dioxide, nitrogen dioxide, suspended particles ($PM_{10}$ and $PM_{2.5}$), lead, benzene, and carbon monoxide, as well as target values for suspended particles ($PM_{2.5}$) and ground-level ozone according to the values from the CAFE Directive [19].

The majority of the local pollution in the area is attributed to the chemical imbalance or to the aerosols, which mostly comes from vehicles or during the combustion processes in industrial plants. Complex chemical processes can alter the physical properties of polluting matter. From a theoretical standpoint, standards for emissions are set in such a way that they take into account the effective concentration level of pollution. Pollution is usually experienced as an economic problem that is tied to production and consumption [20,21]. However, it is important to determine the right values of pollutants in the environment which can cause harm.

Based on the concentration values of the conducted research, it is possible to establish a comprehensive review of the quality of air, that is, air pollution in multiple park locations in the city, and to enable classification of locations according to the scale, which is regulated by the regulation of conditions for monitoring and demands for air quality [18]. Research showed the current state of green parks, whereas the results of this analysis can be used to propose upgrading systems of monitoring in parks and a proposal for landscaping and further air-quality monitoring. Landscaping projects are a special group of air pollution prevention measures in areas where air pollution occurs. Raising green areas in the form of parks, tree lines, hedges, or lawns greatly improves the air quality in the city. Resistant trees

constantly generate new amounts of oxygen, consume harmful carbon dioxide, absorb soot and dust particles, and absorb large amounts of solar radiation with their green canopies, which lowers the temperature and creates more favorable living conditions [22].

The main goal of the paper was to point out the need for air-quality monitoring in parks as the first indicator of air pollution.

This paper focuses on two separate but interrelated goals. First, the primary goal is to determine the current state of concentration levels of $PM_{2.5/10}$ and $NO_2$ in the same locations in the city parks with a laboratory inter-comparison of two sensor methods: stationary and mobile. The inter-comparison has a task to confirm the Aeroqual Ltd. instrument quality of operation. As a secondary objective, this paper will also show the statistical dependence of variable $NO_2$ and microclimate parameters ($T$ °C and RH%) and $PM_{2.5/10}$ using simple linear regression analysis. The paper should also point out the need to monitor park air pollution as a warning about the danger to human health.

The paper is structured as follows: The introduction points out the need, importance and aspects of research, citing the literature. Then, the authors present the methodology: the locations, sampling method, and quantitative control of measurements. In the following sections, the authors present the results of the research and discussion of the results. The final section provides concluding remarks.

## 2. Materials and Methods

### 2.1. Sampling Site

This study has been expanded to three green parks (GP 1–3) in relation to our previous conducted research [23,24]. The selected parks are located in the city of Novi Sad, Serbia.

Limanski (Liman) park (GP1). The largest park in the city is located in a part of the city called Liman 3 and covers an area of 12.9 ha. It is surrounded by large roads on two sides, such as one of the more frequented boulevards and a bridge over the Danube, while the rest of the park is located in a quieter part of the neighborhood [25].

Futoški park (GP2). Futoški Park was opened in the first decade of the 20th century. It covers an area of more than 8 hectares. Futoški Park, positioned in the wider city center, represents a green garden of Novi Sad that has somehow managed to remain hidden in between two noisy streets, which lead to a suburb of Novi Sad. There are also several geothermal springs. The park is classified as a second-level category protected area. Its maintenance and protection are entrusted to the PE City Greenery [25].

Dunavski (Danube) park (GP3). The prettiest and most popular public park in Novi Sad, is located in the very city center and covers more than 33,000 square meters. The park is under the protection of Institute for Nature Conservation of Serbia. It is located near the Danube. On one hand, it is bordered by a boulevard that leads to a bridge over the river Danube, while at the back of the park there are quieter streets and a pedestrian zone [25]. By the Decree on Nature Protection of the III degree, this natural asset of great importance in the II category was declared a Monument of Nature.

Three sampling points (A, B, and C) were selected in each green park. Sampling points A and B are located on the edge of the parks, whereas sampling points C are near the exit of the parks. The main criteria for the selection of these sampling points are the previously conducted research in a similar outdoor environment [23,24].

### 2.2. Particulate Matter (PM) Measurement

Five-day measurements of $PM_{2.5/10}$ were carried out using, the Aeroqual Series 500 (Aeroqual Limited, Auckland, New Zealand) instrument with exchangeable sensor heads [23,24,26]. Aeroqual's low-cost handheld air-quality monitors enable accurate real-time surveying of common pollutants using an ultra-portable device. They are applicable for indoor and outdoor environments. They are used mainly for indicative measurements to point to the need for more detailed research. The disadvantage is that the rechargeable lithium battery allows operation for only up to 8 h [26].

The working principle of a sensor head is to use active sensor technology with an internal fan to draw air through the gas-sensitive sensor at a certain flow rate for accurate gas detection. The fan pulls a stream of air past the sensor every 60 s, providing a new reading of $PM_{2.5/10}$ levels and results in a 60 s response time.

LPC technology, a laser particle counter for measuring particles (PM), uses optimized signal processing courtesy of low-noise electronics with interference correction algorithms.

LPC sensor head has the following characteristics: The measuring range is 0.000–1000 mg/m$^3$, with a minimum limit of detection and a resolution of 0.001 mg/m$^3$. Accuracy is $\pm$(0.002 mg/m$^3$ + 15% of reading). The sensor head can operate without condensation at temperatures ranging from 0 to 40 degrees Celsius and air humidity levels ranging from 0 to 90%.

Daily gas sampling is divided into two time intervals: in the morning (from 10 to 12 am) and at end of the working day (from 4 to 6 pm). Each time interval included 30 measurements in the range of two minutes [23]. Simultaneously, microclimatic parameters were measured with a Mannix DLAF-8000 mini airflow meter, which is a high-precision thin-film capacitance humidity sensor with a fast response to humidity changes. The standard type K (NiCr-NiAl) thermocouple input jack is suitable for all kinds of type K probes. The built-in microprocessor circuit assures excellent performance and accuracy. The range of airflow is 0.01–30 m/s with a resolution of 0.01 m/s and an accuracy of $\pm$3%. The range of humidity is 10–95% with a resolution of 0.1% RH and an accuracy of <70% RH: $\pm$4%RH. The range of temperature is 0–50 °C with a resolution of 0.1 °C and an accuracy of $\pm$1 °C [27,28].

### 2.3. Quantitative Control of Air Quality Using Nonstandard Methods

According to the Law on air quality, Article 8 [29], rating the air quality is performed for the following pollutants: sulfur dioxide, nitrogen dioxide, suspended particles ($PM_{10}$, $PM_{2.5}$), lead, benzene, carbon monoxide, ground-level ozone, arsenic, cadmium, nickel, and benzopyrene. Evaluation of air quality, based on the measured concentrations of polluting matters in the air is performed by applying the criteria for evaluation in accordance with the Regulation on conditions for monitoring and requirements for air quality [29] shown in Table 1.

**Table 1.** Limit values of parameters for protection against human health, according to the Regulation on conditions for monitoring and air quality requirements (2013b).

| Polluting Matter | Averaging Period | Emission Limit Values (ELV) | Can't Be Exceeded More than 10 Times In One Year | Emission Tolerance Value (ETV) (ELV + Limit of Tolerance) | | | | | | Lower Limit of Evaluation | Upper Limit of Evaluation |
|---|---|---|---|---|---|---|---|---|---|---|---|
| | | | | 2011 | 2012 | 2013 | 2014 | 2015 | 2016 | | |
| Nitrogen dioxide NO$_2$ (µg/m$^3$) | 1 h | 150 | 18X | 225 | 217.5 | 210 | 202.5 | 195 | 187.5 | 75 | 105 |
| | 24 h | 85 | - | 125 | 121 | 117 | 113 | 109 | 105 | - | - |
| | Yerly | 40 | - | 60 | 58 | 56 | 54 | 52 | 50 | 26 | 32 |
| Suspended particles, PM$_{10}$ (µg/m$^3$) | 24 h | 50 | 35X | 75 | 70 | 65 | 60 | 55 | 50 | 25 | 35 |
| | Yerly | 40 | - | 48 | 46.4 | 44.8 | 43.2 | 41.6 | 40 | 20 | 28 |
| Suspended particles, PM$_{2.5}$ (µg/m$^3$) | Yerly | 25 | - | 30 | 30 | 29.3 | 28.5 | 27.8 | 27.1 | 12.5 | 17.5 |

To determine the variation of polluting substance emissions, such as NO$_2$, PM$_{2.5}$, and PM$_{10}$, the first step in this inter-laboratory comparison was to collect all relevant data, which are related to:

- Meteorological parameters (temperature, relative humidity);

- Quality of air (concentrations of $NO_2$, $PM_{2.5}$ and $PM_{10}$).

For the chosen pollutants, determining the dispersion of the polluting matter in the air represents the main goal of the possible foresight of air pollution, which is primarily conditioned by traffic conditions with consideration for meteorological conditions. Formerly, in literature, the dispersion of pollutants was considered mostly on the macro level (whole country) or on the mezzo level (city) of air pollution, while lately the focus has been shifted more to the micro level (for example, a certain locality) [30]. Hence, for this purpose, the results that were used were the results of measuring air pollution at a known location.

### 2.3.1. Result Processing

The results are expressed by statistical values; that is, an analysis of the average value is performed, and standard deviation results can be converted into the next statistical parameters according to ISO 13528 [31]:

$$z - score, \; z = (X - T)/\sigma, \tag{1}$$

where $X$ is the measured value, $T$ is the reference value, and $\sigma$ is the reproducibility standard deviation.

With a representation such as this, it is possible to compare the obtained values in a simple way. The calculated $z$-score = 0 represents an ideal value. Satisfactory values of the $z$-score are in an acceptable range from $-2$ to 2. Values that are in the range from 2 to 3 and from $-2$ to $-3$ are considered debatable, while values that are outside of this interval are considered unacceptable and demand a reassessment, finding, and removal of the cause of deviation.

### 2.3.2. Inter-Laboratory Comparison of Quantitative Control of the Quality of Air with Nonstandard SENSOR Methods

The air from locations on the territory of Novi Sad (center), on 27 March 2020 was examined by two laboratories

1.  Laboratory for industrial (applied) electronics, University of Novi Sad, Faculty of Technical Sciences, Department of Power, Electronics and Telecommunication Engineering, Chair for electronics
2.  Laboratory for ecology and environmental protection, University Business Academy in Novi Sad, Faculty of Economics and Engineering Management in Novi Sad.

For the reference value ($T$), data were taken from the results of a relevant stationary measuring station in Novi Sad on 27 March 2020.

$NO_2$ = 13.1 µg/m³; $PM_{2.5}$ = 43 µg/m³; $PM_{10}$ = 88 µg/m³.

Environmental conditions were: $T$ = 9.2 °C; H = 66.3%; p = 969.1 kPa.

Evaluation of the results was performed based on the obtained values of $z$-score.

The results of the inter-laboratory comparisons (Table 2) enable the identification of possible problems and verification work when the methods from the manufacturer's manual are used. Comparative testing with the sensor methods, stationary and mobile, within the Laboratory for ecology and environmental protection and the renowned Laboratory for industrial (applied) electronics has established that the measured values of polluted gases and particle pollution stack up nicely. Given that the satisfactory values of $z$-score are from $-2$ to 2, it can be considered that the method of quantitative control of air quality—with nonstandard sensor methods, sensors GSE (Gas Sensitive Electrochemical) and LPC (Laser Particle Counter) as specified by Aeroqual Ltd.—provided a constant quality during operation.

**Table 2.** Results of the inter-laboratory comparison.

| Laboratory | Average Value of 30 Measurements | | | |
|---|---|---|---|---|
| | $NO_2$ µg/m$^3$ | $PM_{2.5}$ µg/m$^3$ | $PM_{10}$ µg/m$^3$ | Measuring Methods |
| Laboratory for industrial (applied) electronics | $\overline{X}$ = 19.17 S = 0.941 Z = 1.20 | $\overline{X}$ = 42.13 S = 6.749 Z = −0.05 | $\overline{X}$ = 60.39 S = 14.443 Z = −1.20 | Sensor methods—remote measuring stations for tracking parameters of the environment |
| Laboratory for ecology and environmental protection | $\overline{X}$ = 19.26 S = 0.655 Z = 1.80 | $\overline{X}$ = 39.60 S = 11.859 Z = −0.11 | $\overline{X}$= 65.50 S = 19.326 Z = −0.80 | Quantitative air quality control—non-standard sensor method, GSE and LPC sensors according to Aeroqual Ltd. specification |

### 3. Results

Measured concentrations of $PM_{2.5/10}$ in Novi Sad parks are in the range of 25–30 µg/m$^3$, which is below the maximum daily value for both particle sizes [23,24]. These values are conditioned by a relatively low temperature of 17–18 °C and humidity of 47–49%, and by an intense wind, which enables the transport of particles over greater distances and leads to their "dilution". Based on these low concentration values of suspended particles ($PM_{2.5/10}$) in the parks of Novi Sad, it can be considered that the air is moderately clean, given that the maximum allowed value was not exceeded for PM particles. With the change in weather conditions, the particle level at all the measuring locations is higher due to the amplified sunshine and the decrease in wind speed. Considering that the measurements were performed in parks, there is a possibility that the health of the people in those parks could be endangered during long-term exposure.

*The Simple Regression Analysis*

The functional relationship between PM as a dependent variable and temperature, humidity, and $NO_2$ oxide content as independent factors was determined using the results of the analysis. The quantitative dependency between the fluctuations of the observable occurrences in reality is best described by this simple model.

A straight line drawn as close to all empirical sites as possible is the functional form of connection [32]. Because the findings of $PM_{2.5}$ and $PM_{10}$ measurements in all parks were identical, the values of all parks were combined to form a single green belt in the city. In earlier investigations [23], we discovered that there was a substantial link between $PM_{2.5}$ concentrations and $PM_{10}$, $NO_2$, $O_3$, CO, air temperature, and air humidity.

In urban environments, air pollutants (such as $SO_2$, $NO_2$, CO, and others) are the most common cause of PM generation, which varies in different parts of the city at different times.

To observe, a basic linear regression analysis was employed, (Figures 1–6):

However, the majority of studies focus solely on the trend of changes in pollutant concentrations over time, with data on the ratios of pollutant concentrations and meteorological circumstances being scarce [15,33].

In Table 3, the equations for the observed dependencies are given. Table 4 shows the obtained coefficient of simple linear correlation between two variables in the sample, or Pearson correlation coefficient r, relative measure of representation of the regression line—coefficient of determination—$R^2$, and values of testing of significance of the regression connection—Student's test—*t*.

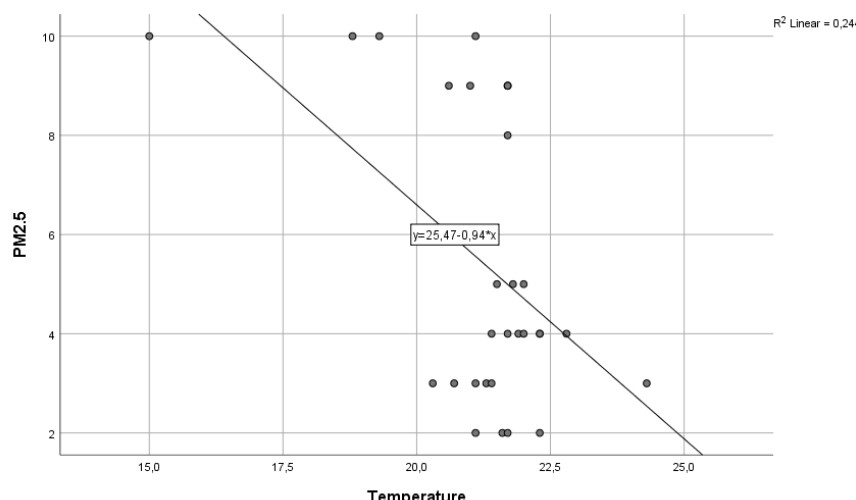

**Figure 1.** A regression line depicting the effect of temperature on PM$_{2.5}$ concentrations.

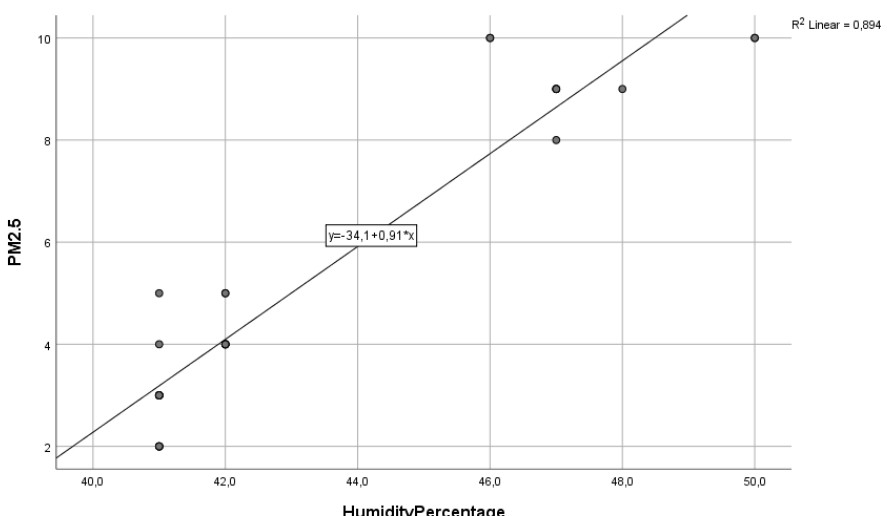

**Figure 2.** A regression line depicting the impact of air humidity on PM$_{2.5}$ levels.

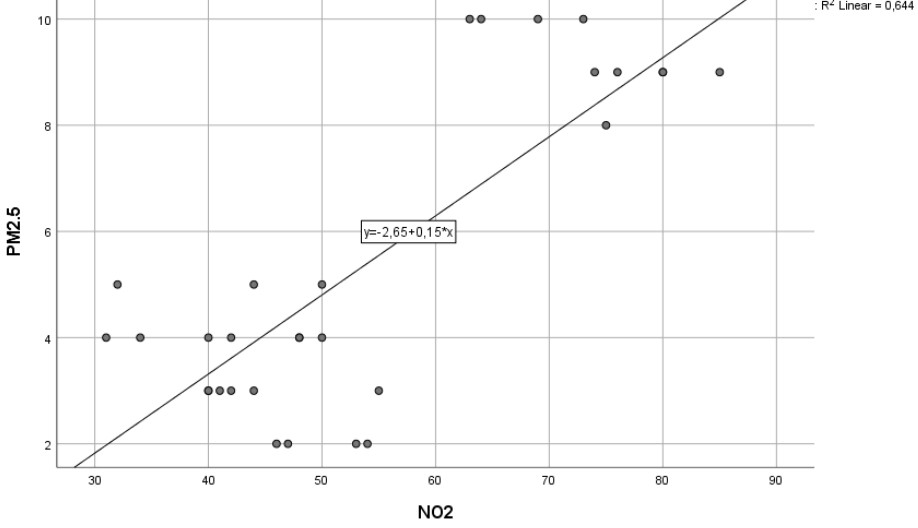

**Figure 3.** A regression line depicting the impact of NO$_2$ concentration on PM$_{2.5}$ levels.

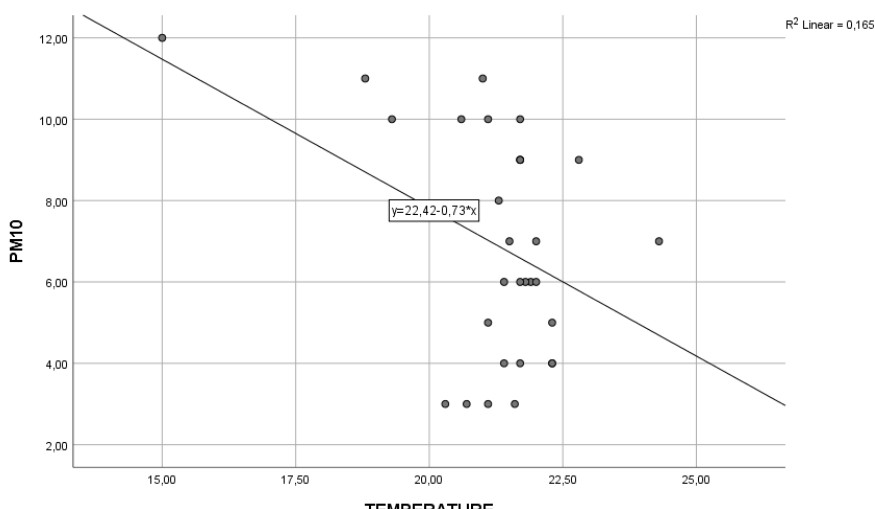

**Figure 4.** A regression line depicting the effect of temperature on $PM_{10}$ concentrations.

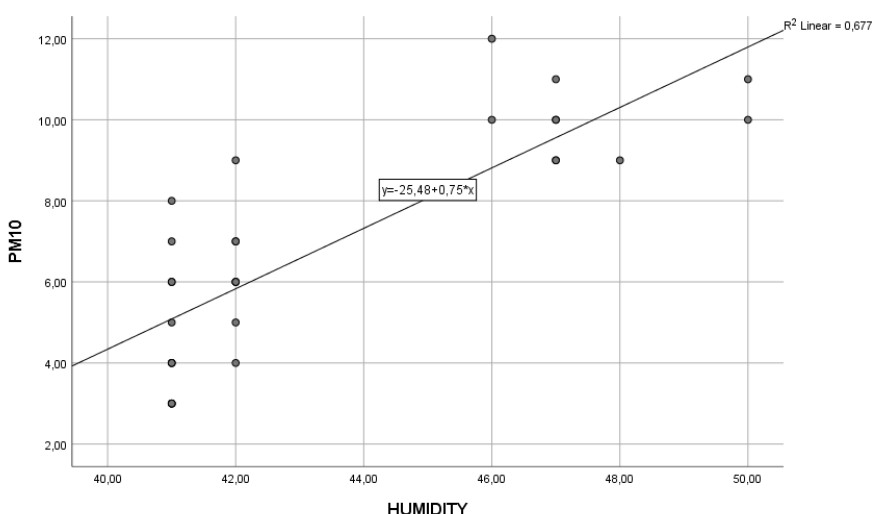

**Figure 5.** A regression line depicting the effect of air humidity on $PM_{10}$ levels.

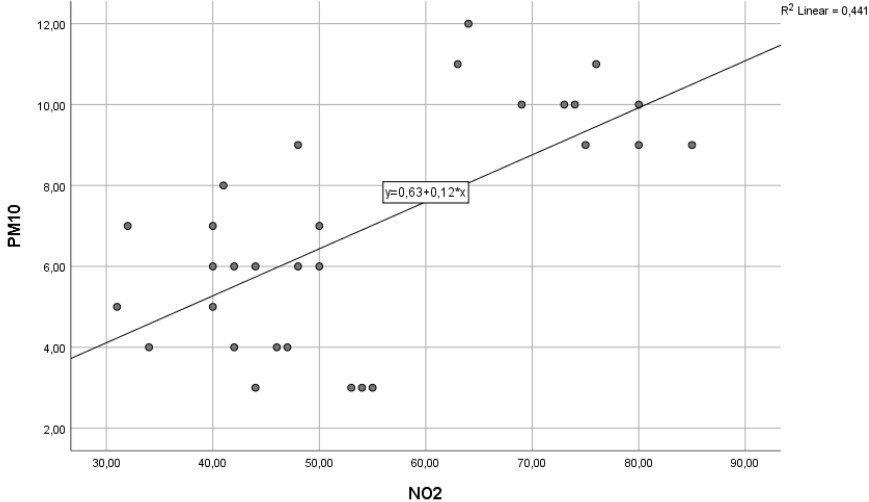

**Figure 6.** The relationship between NO2 concentration and PM10 concentration.

**Table 3.** Equations of the simple linear regression of influence of temperature, air humidity and concentration of $NO_2$ on the concentration of $PM_{2.5}$ and $PM_{10}$ particles.

| Observed Dependencies | Equation |
|---|---|
| Influence of $T$ °C $PM_{2.5}$ concentrations | y = 25.47 − 0.94x |
| Influence of $NO_2$ oxide concentration on $PM_{2.5}$ concentrations | y = 2.65 + 0.15x |
| Influence of air humidity on $PM_{2.5}$ concentrations | y = −34.10 + 0.91x |
| Influence of $T$ °C on $PM_{10}$ concentrations | y = 22.42 − 0.73x |
| Influence of $NO_2$ oxide concentration on $PM_{10}$ concentrations | y = 0.63 + 0.12x |
| Influence of air humidity on $PM_{10}$ concentrations | y = −25.48 + 0.75x |

**Table 4.** Coefficients of simple linear regression analysis of the influence on $PM_{2.5}$ and $PM_{10}$ concentration.

| Observed Dependencies | R | $R^2$ | t |
|---|---|---|---|
| Influence of $T$ °C $PM_{2.5}$ concentration | 0.494 | 0.244 | −3.009 |
| Influence of $NO_2$ oxide concentration on $PM_{2.5}$ concentration | 0.802 | 0.644 | −7.110 |
| Influence of air humidity on $PM_{2.5}$ concentration | 0.945 | 0.894 | 15.330 |
| Influence of $T$ °C on $PM_{10}$ concentration | 0.406 | 0.165 | −2.350 |
| Influence of $NO_2$ oxide concentration on $PM_{10}$ concentration | 0.664 | 0.441 | 4.703 |
| Influence of air humidity on $PM_{10}$ concentration | 0.823 | 0.677 | 58.740 |

The obtained diagrams (Figures 1–6) and equations (Table 3) show that there is a quantitative match:

- The dependence of the concentration of $PM_{2.5}$ and $PM_{10}$ particles on $NO_2$ levels and humidity is direct and there is a markedly linear correlation because the coefficient of determination is $R^2 = 0.644$ and $R^2 = 0.894$; that is, $R^2 = 0.664$ and $R^2 = 0.823$;
- The dependence of $PM_{2.5}$ and $PM_{10}$ particles on temperature is a linear and inverse connection, and it is unexpressed because the coefficient of determination is $R^2 = 0.244$ and $R^2 = 0.1605$.

## 4. Discussion

According to the obtained coefficients of simple linear correlation between the two variables in the sample or the Pearson coefficient of correlation r = 0.945 for pollutant particles $PM_{2.5}$ and r = 0.823 for $PM_{10}$ particles, the coefficient of determination $R^2 = 0.893$ for $PM_{2.5}$ and $R^2 = 0.677$ for $PM_{10}$, and the calculated statistics of the Student's test $T = 15.330$ for $PM_{2.5}$ and $T = 58.740$ for $PM_{10}$ (maximum value), air humidity has the biggest influence on the concentration of $PM_{2.5}$ and $PM_{10}$ particles. The obtained result shows us that 89.3% and 67.7%, respectively, of the total variation of the concentration of $PM_{2.5}$ and $PM_{10}$ particles, is determined by the independent variable X; that is, the change in air humidity. The rest, 10.7% and 32.3%, respectively, is not explained by the regression equation; that is, it is under the influence of unidentified factors.

It must also be mentioned that measured concentrations of particulate pollution during 2020 in all three city parks in Novi Sad, for $PM_{10}$, were in the range of 9–12 μg/m$^3$ and for $PM_{2.5}$, in the range of 2–10 μg/m$^3$ at air temperatures of 15–23 °C, and humidity of 46–50% with medium-strength wind. The measured concentrations of particulate pollution in 2019 in all three Novi Sad parks were the same and amounted to about 30 μg/m$^3$ [23].

Comparing the measured concentration values of $PM_{2.5/10}$ before and during the COVID-19 pandemic, it is concluded that all measured values in 2020 were drastically lower. From the difference between the measured maximum and minimum values of $PM_{2.5/10}$ for 2019 and 2020, a percentage reduction of particulate pollution was obtained during the two-year measurement in the parks of Novi Sad: for GP1 91% and 82%; for GP2 70% and 69%; and for GP3 80% and 84% for $PM_{2.5}$ and $PM_{10}$, respectively [24]. These comparative data have a large disproportion for 2019 and 2020 and indicate a noticeable decrease in airborne particles in 2020, which is a consequence of the extraordinary pandemic–epidemic situation that affected the whole world, but also the parks locally.

## 5. Conclusions

An air quality examination in the city parks of Novi Sad, at three different locations, confirmed the interdependence of the present air pollutants and their interaction with microclimate conditions in real systems. The best and most effective approach is simultaneous and comprehensive measuring of meteorological parameters, and concentrations of particulate matter in the air at multiple locations in parks. The quantitative control of air quality with inter-laboratory comparison confirmed that the non-standard Aeroqual Ltd. method, using GSE sensors (gas-sensitive electrochemical) and LPC (laser particle counter), provides a constant quality of operations. The z-score, the measure of comparing the results, has values from $-2$ to $2$, which is satisfactory. It was also confirmed by using simple linear regression that there are linear correlations between the two variables (PM and $NO_2$) in the sample and that the air humidity has the largest influence on the concentration of $PM_{2.5}$ and $PM_{10}$ particles, which explains the total variations of the concentration of $PM_{2.5}$ and $PM_{10}$ particles of 89.3% and 67.7%, respectively, with the change in air humidity.

Based on the results of the monitoring, it is possible to establish a comprehensive review of air quality and pollution at several park locations in the city and enable the classification of locations in accordance with the scale prescribed by the Regulation on monitoring conditions and air-quality requirements. The outcome of the current state of air pollution in parks could be used to inform public health experts and policymakers about the benefits of providing access to green spaces to citizens, as well as to understand the potential of urban green spaces in health promotion.

Research results enable the comparison of air pollution at locations of regular measurements and air in green spaces and parks, as well as pollution in various different weather conditions.

Finally, the results of the conducted monitoring should initiate the first plans for controlling pollution in city parks. Long-term results would be applicable in transforming society into a green economy, improving air quality, promoting innovative approaches, and promoting good practices in resolving problems of pollution on a local level.

**Author Contributions:** Conceptualization, J.S.K.; methodology, J.S.K. and V.M.R.; mathematical processing of the measurement results, S.M.A.; validation, J.S.K., D.D.S. and V.M.R.; formal analysis, J.S.K., A.M.Č., D.D.S., V.M.R.; investigation, J.S.K., V.M.R. and S.M.A.; resources, D.D.S.; data curation, V.M.R.; writing—original draft preparation, J.S.K., A.M.Č., S.M.A.; writing—review and editing, J.S.K., V.M.R.; visualization, V.M.R.; supervision, J.S.K. and V.M.R.; All authors have read and agreed to the published version of the manuscript.

**Funding:** This work is funded by the Ministry of Education, Science and Technological Development of the Republic of Serbia: "Innovative Scientific and Artistic Research from the Faculty of Technical Sciences Activity Domain", ID: 200156.

**Institutional Review Board Statement:** Not applicable.

**Informed Consent Statement:** Not applicable.

**Data Availability Statement:** Not applicable.

**Conflicts of Interest:** The authors declare no conflict of interest.

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
