# Peer review of "Contribution of Particles to Air Pollution in Green Parks"

_sustainability, doi:10.3390/su14063534_

Round 1

Reviewer 1 Report

Dear author(s),

The idea of this study is interesting and the procedures and steps of the research have been done logically. There are some major points that may improve the quality of the manuscript, and the soundness of sentences needs serious improvement. There are some major points that considering them and modifying the text based on them, will improve the quality of the manuscript. The points are listed as the following.

  • The Abstract section should be rewritten. There is no coherence and cohesion among the sentences. After the first two sentence, that are general speaking about parks and pollutions, the text drops to the parks of Novi Sad city, which is not appropriate. Moreover there are not enough explanations about the methods, applications, tools, and also the main numerical results obtained in this study. Therefore, it is recommended to rewrite the Abstract section precisely.
  • Line 39: Mentioning the “preventive measures” needs to be supported by sentences to explain the reason of it.
  • In the Introduction and Materials and methods sections there are lots of extra mentioning and explanations about the names and details of the parks in the city. A scientific article should not be considered as an advertisement brochure. This part should be summarized. Also there is no need to mention the name of all the parks in the city that are not related to this study.
  • Line 80: It should be mentioned that according to which source Novi Sad used to be the greenest city.
  • Line 97: In the first time of using any acronym or abbreviation, it should be mentioned in expanded format. For instance the “PM” is used in lines 97 and 99 etc. but it is mentioned for the first time in line 187.
  • The content of lines 119 to 122 has been repeated in line 404 to 407. Please modify the manuscript and clean it from such repetitions.
  • The last paragraph of the Introduction section should be related to the structure and sectioning of the article, which is missed in the manuscript. It is suggested to add such a paragraph.
  • In section 2.1, the authors mentioned about their previous studies but they did not mention the significant differenced among these studies.
  • Line 207: Both “nitrogen dioxide and nitrogen oxides” have been mentioned. Please recheck whether it is meaningful or not.
  • The first paragraph of the Results and discussion section needs to be removed from this section and added to Sections 1 or 2.
  • From line 310 to 323, the authors discussed about the linear regression model, which is the simplest and the most primitive regression models. Surely, it does not need to be explained in that details in a scientific article. It is recommended to be summarized.
  • In the last paragraph of the Discussion and Conclusion section, it is recommended to mention about the limitation of this study, and the potential areas for the future studies in this field.
  • In general, the referencing of the manuscript needs serious improvement. The authors should bring the reference(s) for the definitions, sentences, etc. that they use from other studies, and must cover more recent studies from the literature. For instance,
    • At the end of the first paragraph of the Introduction section, there is a lack of at least a reference.
    • Line 80: “(Jovanović and Gaudenji 2005), should be corrected to (Jovanović and Gaudenji, 2009), according to the related reference in that section.
    • Line 106: By switching the references in lines 483 and 486, the reference in this line should be (Official Gazette of RS 2013a) as it is used first.
    • Line 109: The “café directive” has been mentioned “Directive café” in the reference list.
    • Line 228: Mentioning such a formula needs a reference.
    • If I am not mistaken, the following reference has not been mentioned in the text. Please recheck it.
      • Miao W, Huang X, Song Y (2017) An economic assessment of the health effects and crop yield losses caused by air pol-479 lution in mainland China, J Environ Sci., 56, 102-113, ISSN 1001-0742, https://doi.org/10.1016/j.jes.2016.08.024. 480
    • The text should be controlled by a professional English writer, as there are mistypes that some of them are mentioned in the following.
      • Line 28: Using “Such as, for example …” is wrong.
      • Line 34: Using sentences like “The main concentrated sources of emissions come from big industrial sites” is not appropriate in a scientific article.
      • Line 57: “Besides that” should be corrected to “Besides”.
      • Line 65: “Environmental” should be started with a small letter as it is in the middle of the sentence.
      • Line 142: “a fresh-water spring” does not seem to be correct.

It is recommended to recheck the whole manuscript according to the above-mentioned English, punctuations, and mistyped problems.

Author Response

Thank you for the suggestions. Corrections can be found in the attached file.

Reviewer 2 Report

The content of the submitted manuscript is good but the presentation way of current form is not fulfilling the journal requirements. Modification is needed to consider for publication.

Figures: Reviewer couldnt receive the figure files / single figure 

Table: Reviewer couldnt receive the table files / single table

  - Title of the paper

The title of the paper looks good but in the same time, it can be modified to represent the manuscript in a better way.

Abstract

-          The abstract is not well written

-          You should include some of the main finding in the abstract section.

Abstract should have a conclusion of the study.

Introduction  

  • The objective of the study is also not clearly mention.
  • Add more on the basic of the problem in the introduction
  • The author should focus mainly on the importance and significance of the study.
  • I suggest the author to demonstrate what does the paper add to the current literature? and what new knowledge is added by this study?

Add the unique of this study compared to other studies discuss the same issue.

-Discus merits and limitations of technique applied.

 Material and Methods

-     The material and method section is too weak in the manuscript and you need to focus on it more.

Result and discussion

  • The presentation fails to discuss the summary, and trying to some of vague reason which is not the explanation.
  • The explanation for the critical analysis is not sufficient, although some of the good points have has been identified.

Conclusion

-          Please rewrite the conclusion with the proper explanation in the R & D.

References

Reference section should be increased with number of recent studies. I would like to suggest to author to include following published article to improve the quality of articles, these are

Characterization, seasonal variation, source apportionment and health risk assessment of black carbon over an urban region of East India (10.1016/j.uclim.2021.100896)

Bioaerosols: Characterization, pathways, sampling strategies, and challenges to geo-environment and health (10.1016/j.gr.2021.07.003).

Other comments:

English editing is needed in some parts of the manuscript.
Abbreviations should be explained before the introduction.

Author Response

(The authors gave the same response as above.)

Round 2

Reviewer 1 Report

Dear respected Authors,

The revised version of the manuscript, entitled "Contribution of particle air pollution in the green parks" has been reviewed. All the comments have been answered patiently and the manuscript has been modified precisely, according to the reviewer’s comments. The needed corrections of mistypes and English polishing of the revised version of the manuscript have been performed too. Therefore, the revised version of the manuscript is worth to be published in the respected Journal of Sustainability.

Warm regards and congratulations

Author Response

The authors would like to thank you for a timely and detailed analyses of our manuscript, as well as for constructive comments that have led to the improvement of our manuscript.